# Bioactive Graphene Quantum Dots Based Polymer Composite for Biomedical Applications

**DOI:** 10.3390/polym14030617

**Published:** 2022-02-05

**Authors:** Seyyed Mojtaba Mousavi, Seyyed Alireza Hashemi, Masoomeh Yari Kalashgrani, Navid Omidifar, Sonia Bahrani, Neralla Vijayakameswara Rao, Aziz Babapoor, Ahmad Gholami, Wei-Hung Chiang

**Affiliations:** 1Department of Chemical Engineering, National Taiwan University of Science and Technology, Taipei City 106335, Taiwan; kempo.smm@gmail.com (S.M.M.); vijayrao@mail.ntust.edu.tw (N.V.R.); 2Nanomaterials and Polymer Nanocomposites Laboratory, School of Engineering, University of British Columbia, Kelowna, BC V1V 1V7, Canada; s.a.hashemi0@gmail.com; 3Biotechnology Research Center, Shiraz University of Medical Sciences, Shiraz 71468-64685, Iran; masoomeh.yari.72@gmail.com (M.Y.K.); S.bahrani22@gmail.com (S.B.); 4Department of Pathology, Shiraz University of Medical Sciences, Shiraz 71468-64685, Iran; omidifar@gmail.com; 5Department of Chemical Engineering, University of Mohaghegh Ardabil, Ardabil 56199-11367, Iran; Babapoor@uma.ac.ir; 6Pharmaceutical Sciences Research Center, Shiraz University of Medical Sciences, Shiraz 71468-64685, Iran

**Keywords:** polymer composites, graphene quantum dots, bioactive, biomedical, synthesis

## Abstract

Today, nanomedicine seeks to develop new polymer composites to overcome current problems in diagnosing and treating common diseases, especially cancer. To achieve this goal, research on polymer composites has expanded so that, in recent years, interdisciplinary collaborations between scientists have been expanding day by day. The synthesis and applications of bioactive GQD-based polymer composites have been investigated in medicine and biomedicine. Bioactive GQD-based polymer composites have a special role as drug delivery carriers. Bioactive GQDs are one of the newcomers to the list of carbon-based nanomaterials. In addition, the antibacterial and anti-diabetic potentials of bioactive GQDs are already known. Due to their highly specific surface properties, π-π aggregation, and hydrophobic interactions, bioactive GQD-based polymer composites have a high drug loading capacity, and, in case of proper correction, can be used as an excellent option for the release of anticancer drugs, gene carriers, biosensors, bioimaging, antibacterial applications, cell culture, and tissue engineering. In this paper, we summarize recent advances in using bioactive GQD-based polymer composites in drug delivery, gene delivery, thermal therapy, thermodynamic therapy, bioimaging, tissue engineering, bioactive GQD synthesis, and GQD green resuscitation, in addition to examining GQD-based polymer composites.

## 1. Introduction

Bioactive materials play an increasingly important role in the biomaterials industry and have wide applications in nanomedicine, drug delivery systems, and biosensors. Among the medical applications of bioactive substances are the continuous development and current status of bioactive substances in medicine [1]. The term graphene was first introduced in 1986 and was created by combining the word graphite with a suffix (n) referring to polycyclic aromatic hydrocarbons. In various fields of science and technology, graphene has changed its status from an unknown substance to a bright star. This is due to the exceptional properties of graphene, including high current density, ballistic transport, chemical inertness, high thermal conductivity, optical permeability, and excellent hydrophobicity at the nanometer scale [2,3]. The first use of functionalized graphene oxide polyethylene glycol as a nan carrier of anticancer drug release was reported in Liu et al. [4]. Sanchez et al. investigated the biological properties of graphene nanomaterials and their toxicity, containing their proper interaction with biomolecules, cells, and tissues based on the number of layers and the dimensions of chemical functionalization [5]. Quantum dots are mineral semiconductor nanocrystals with a diameter of 1 to 10 nanometers that emit light after excitation and typically consist of 100 to 10,000 atoms. Their small size makes them unique compared to macrocrystalline materials [6]. In many branches of science, GDs with unique properties have been used. The use of these GDs in pharmaceutical and medical technology is also evolving [7,8]. The remarkable features of quantum dots that have made them a probe for targeted drug delivery and therapeutic fields are their rich surface area and optical properties.

The cells or tissues in question are targeted by QDs, combined with ligands. Recently, the use of QDs for in vivo applications has become more prominent with advances in the production of biocompatible QDs [9]. QDs can also lead to good solubility in various solvents, such as aqueous buffers, when in the form of a semiconductor core with a coating (as a shell) and a cap [10]. Luminescent nanocrystals, known as QDs, are used for carriers or probes in medical applications due to their rich surface chemistry, such as in drug delivery [11,12,13,14], imaging, and treatments [15]. Graphene quantum dots (GQDs) are graphene-based nanomaterials [16]. In other words, GQDs that are the product of the chemical oxidation process of carbonaceous materials can be considered extremely small derivatives of graphene oxide which contain a significant number of oxygenated functional groups, such as hydroxyl and carboxyl [17,18]. The attention of many researchers has been drawn to GQDs due to their excellent properties, such as low toxicity, good water solubility, electrical conductivity, biocompatibility, stable photoluminescence, and surface to volume ratio, and also to the application of GQDs in bioimmography, sensors, tissue engineering, photocatalysis, and energy conversion [19]. Photoluminescence is one of the main properties of GQDs that sometimes explain the diversity of their biomedical applications. However, the photoluminescence mechanisms of GQDs are not well known because their structure is complex and their surface functionalities are still unknown. Therefore, achieving optimal optical properties that affect biomedical applications is challenging.

So far, no mechanism has comprehensively explained the photoluminescence phenomenon of GQDs, limiting the regulation of their optical properties. However, several mechanisms have been proposed, including the quantum confinement effect, the emission of surface states, molecular fluorophores, the molecular diffusion of polycyclic aromatic hydrocarbons, self-trapped exciton emissions, the surface dipole emission center, the aggregate emission center, the multiple dissolution center, the relaxation of slowed solvent center, and a solvatochromic shift. Each mechanism only partially explains the photoluminescence phenomenon of GQDs. It has been suggested that bioactive compounds can specifically improve photoluminescence. The polymerization of GQD structures can also increase the biological effects of GQDs.

GQDs are typically reported to be between 3 and 20 nanometers in size, with the largest reported size being 60 nanometers. Structurally, a GQD is crystalline and has one or more layers of graphene [20,21]. GQDs can interact with biomolecules, so they can be used for drug and gene delivery by making appropriate surface changes [22]. For example, Huang et al. designed GQD nanocarriers that were able to simultaneously use specific drug delivery, imaging, and chemotherapy for cancer [23]. Apul et al. reported the synthesis of GQDs using citric acid and dicyandiamide as the starting material at 180 °C for 180 min. The synthesized GQDs showed a quantum efficiency of 36.5% and high optical stability [24]. In other studies, GQD synthesis was reported as a bottom-up method based on the degree of carbonization of citric acid [25].

The term composite (multiplayer or composite material) refers to a material that consists of two phases, a matrix, and a reinforcement, and the second phase is used at a maximum of 0%. A matrix composition with fibers (or reinforcing material) below 0% is referred to as a composite. Polymers in which fillers (in the size range from 1 to 100 nm) are homogeneously reinforced are called polymer nanocomposites (PNCs). In these nanomaterials, fillers are specified in at least one dimension less than 100 nm (<100 nm). Based on the number of dimensions that are outside the nanometer range, nanofillers can be classified into three categories: (1) Spherical (zero-dimensional), (2) layered (one-dimensional), and (3) fibrous and tubular (two-dimensional) [26,27,28,29,30,31].

There is a better interaction between the polymer matrix and the filler in polymer nanocomposites than conventional composites. The uniform distribution of nanoparticles in the polymer matrix increases the contact surface made up of the matrix and nanoparticles and improves its mechanical, thermal, and obstruction properties. The smaller the size of the nanoparticles, the more difficult it is to distribute them in the polymer matrix because the nanoparticles are very unstable and have a high tendency to aggregate or clump. The clumping of the nanoparticles is a weakness of the polymer matrix. Mineral nanoparticles with petite particle sizes make minimal changes, as in the case of polymeric materials. If properly designed and formulated, they can improve the polymer’s thermal and mechanical barriers and its flammability properties [32]. Surface charge density, doping status, the degree of surface oxidation, or inks with polymers are among the properties of GQDs that can cause different interactive behaviors with biomolecules and which are dramatically different in additional others [33]. The elastomer matrix is combined with filler properties such as nanoparticles or graphene by flexible and durable polymer composites. The potential application of graphene sheets [34,35,36] as fillers is possible due to their outstanding optical, electrical, and mechanical properties in the context of synthesizing new nanocomposites based on polymers [37,38,39,40]. 

One of the most important aspects of polymer composites when it comes to obtaining lightweight materials with superior performance is the excellent mechanical and electrical properties that graphene imparts to polymer matrices. The outstanding properties that GQD-polymer composites possess with the introduction of only small amounts of nanofillers into the polymer matrix are still being considered [41,42,43]. Thus, research groups have made significant efforts to prepare GQD-polymer composites and study their properties [44,45,46,47,48,49,50]. Information on GQD-polymer composites, including the small size of GQD particles, their atomic rough surfaces, and the dispersibility in polymers have been obtained through simultaneous studies of their mechanical properties and thermal conductivity [51]. Graphene-bound biomacromolecules can improve the bioactivity and biocompatibility of advanced GQD-based biocomposites and nanoplatforms [52,53]. QD-based polymer composites on graphene base surfaces can provide significant biocompatibility and bioactivity, with many potential biomedical and medical applications [54,55,56,57]. Recently, graphene and bioactive GQD based polymer composites have been widely used in medicine for drug delivery, cancer treatment, tissue engineering, phototherapy, and stem cell and biosensor separation [58,59].

This study aimed to present advances in bioactive graphene quantum dot-based polymer composites, focusing on advances in bioactive GQD-based synthesis methods and highlighting future work areas in this field. These methods are based on techniques that rely on top-down or bottom-up processes: bioactive GQD synthesis, GQD green reduction, and GQD-based polymer composites. Finally, the applications of bioactive GQD-based polymer composites, including drug delivery, gene delivery, heat treatment, thermodynamic therapy, bioimaging, and tissue engineering, are discussed. The contents of this review are shown in Figure 1.

## 2. Bioactivation

Bioactive compound is a term commonly used to refer to secondary metabolites produced by organisms which are often not necessary for the growth and survival of organisms and, unlike the primary metabolites and vital macromolecules, are not the basis of the primary processes of the organism and do not play an important role. In general, secondary metabolites in bacteria result from specific conditions, such as limited food sources, and are produced during the idiophase period of the organism’s life. Most of these compounds support the organism in specific environmental conditions (such as competition in the ecosystem). Zahner has proposed the most convincing theory for producing secondary metabolites, which he refers to as secondary metabolite evolutionary compounds. If a metabolite does not affect any stage in the differentiation of an organism (morphogenesis, movement, regulation) and is produced for specific purposes in organisms, it is called a secondary metabolite. Many of these compounds have an antifeedant function in certain organisms, sex absorbers, antibiotic agents [60,61]. 

The first report on the production of biologically active substances from marine bacteria was made by Zobell and Rosenfeld in the context of antibiotic production. Since then, several reports have been received in this regard. When it comes to biological compounds, 16,000 have been isolated from marine microorganisms, including antibacterial, antiviral, and antitumor compounds [62,63].

### 2.1. Polymers Containing Bioactivation

The stabilization of bioactive compounds to produce the functional properties of polymers has developed extensively over the past decade. This method is used in packing industries of foodstuffs and pharmaceuticals. Various bioactive compounds can be used in the stabilization process, such as proteins, peptides, antimicrobial compounds, but most are enzymes which stabilize the surface of polymers. The first step is to stabilize, design, or select a suitable polymer. The mechanical properties considered include elasticity, strength, transparency, and degradability. Stabilization operations can be performed on the surface or inside the polymer. Therefore, the polymer surface must be ready to accept biomolecules. Creating the ability to obtain and bind bioactive compounds in polymers is referred to a special operation called producing functional properties [64,65]. The stabilization of biomolecules and bioactive compounds in polymers is performed using the following methods: (1) Physical absorption, (2) entrapment, (3) covalent immobilisation, and (4) affinity immobilisation [66].

### 2.2. Methods of Bioactivation

#### 2.2.1. Bioactive Carbon Sources

For biomedical applications such as MSC-based therapy, the development of bioactive materials is desirable when it comes to integrating efficient differentiation and traceable properties in a biocompatible manner. A new type of one-step hydrothermal method makes bioactive carbon dots (CD) [67]. Recently, a bottom-up synthetic strategy has been used to design and fabricate multifunctional CDs with the complex introduction of bioactive precursors, including small molecules, nucleic acids, and proteins [68,69]. Compared to traditional citric acid-based CDs, these bioactive CDs can expand their applications against cancer and pathogens and have different pharmacological activities [70,71,72]. In both in vitro and in vivo studies, bioactive CDs have shown improved anti-cancer performance when compared with small free molecules [73]. Therefore, dual-function precursors have not yet been developed to obtain bioactive CDs.

#### 2.2.2. Biomass-Waste Derived GQD

When it comes to the scalable production of GQDs, biomass has been used as a source of green, natural, cheap, sustainable, and renewable carbon. Since biomass is abundant and inexpensive, and there is no published paper on the total cost of GQD materials, it is therefore expected that the price of biomass-derived precursors (waste) is much lower than other precursors (graphite, carbon fiber, CNTs, citric acid, glucose). GQD production with a product yield comparable to expensive graphene-based precursors is possible using different types of biomass, such as plant leaves, grass, rice husk, coffee grounds, and charcoal [74,75,76,77,78]. The quantum efficiency of GQDs derived from biomass is superior to graphene derivatives. According to reports, green production is possible through plant leaves, and GQD production is also done without inactivators, reducing agents, oxidants, or organic solvents [75].

#### 2.2.3. Biologically Active Agents

Bioactive compounds and diagnostic probes, known as biologically active agents, effectively load drug delivery systems based on enzyme-responsive polymers via covalent or physical encapsulation. The isolation of enzyme-responsive polymers for subsequent therapeutic or diagnostic activities is performed by target enzymes to expose or activate biologically active agents. To achieve drug release/activation and physicochemical alternation, enzyme-responsive drug delivery systems can be designed to take advantage of high enzymatic properties [79]. The separation of bioactive agents after enzyme-induced cleavage is one of the direct release/activation methods, which leads to the explosive release of therapeutic cargoes or the activation of extinguished/protected functional agents [80,81,82].

### 2.3. Roles of Bioactivation

Metabolic activation or bioactivation is a relatively inactive organic compound produced by reactive electrophilic intermediates, including free radicals. Bioactivation plays a significant role in hepatotoxicity, dermal ADRs, blood dyscrasias, and drug-induced toxicities [83,84]. The formation of stable/inactive or reactive metabolites is accomplished by the drug’s metabolism, enabling them to interact with cellular structures and functions, which indicates the role of bioactivation in drug-induced hepatotoxicity. Hepatotoxicity is induced by the reaction of electrophilic metabolites with nucleophilic sites in macromolecules either through the metabolic bioactivation of drugs in the formation of covalent compounds or through oxidative damage caused by ROS production [85].

### 2.4. Design of Bioactivation

Electrophiles have attracted the most attention concerning bioactivation in drug design because they are easier to trap and their chemistry is rationalized [86]. Thus, ambiguity in understanding subsequent covalent bonding can be linked with toxicities [87]. In contrast, the initiation of downstream cellular events associated with oxidative stress is better understood through the formation of free radicals. Most of the structural features associated with bioactivation during drug design are related to active aromatic systems (usually aniline derivatives). Assessing the risks involved in developing compounds with structural alerts and the placing bioactivation data during drug discovery are critical issues in the context of bioactivation and drug design [88].

### 2.5. Method of Bioactive GQD Synthesis

The synthesis method plays a significant role in the properties and application of bioactive GQDs. Therefore, the production of bioactive GQDs must be adjusted during and after their production [68]. Chemical, physical, and electrochemical techniques, including various methods, have been used to generate bioactive GQDs [89]. Bioactive GQDs are synthesized top-down and bottom-up (Figure 2). The different methods for synthesizing bioactive GQDs used in biomedicine are listed in Table 1. In the top-down synthesis method, carbon mass decomposition in nanoparticles takes place via chemical and physical processes such as acid oxide, electrochemistry, and hydrothermal processes [90,91,92]. Using a variety of organic compounds under carbonization conditions, GQDs are prepared in a bottom-up method [93]. Among the advantages of bottom-up synthesis are the regulation of physical properties, the use of various organic compounds as raw materials, and the structural properties of GQDs. Processes including stepwise solution chemistry, microwaves, hydrothermal processes, ultrasonic chemistry, acid oxidation, and electrochemistry make up the various physical and chemical methods used to prepare GQDs [94,95,96]. The restriction of access to the mass production of bioactive GQDs using this method is due to the high cost spent on raw materials in the method. Recently, due to the simplicity, cost-effectiveness, and one-step synthesis route method of the hydrothermal method, this method has been considered [25]. The hydrothermal method [97,98,99], microwave method [100,101,102], electrochemical oxidation method [103,104,105], the ultrasonic method [103,106], and biosynthesis methods [107,108] are among the methods used for bioactive GQDs synthesis based on top-down and bottom-up processes.

#### 2.5.1. Hydrothermal Method

A fast and straightforward way to prepare bioactive GQDs is the hydrothermal method. The hydrothermal method is a one-step method in which the organic precursor is heated in a Teflon tube to reach high temperatures and pressures. As a result of high temperatures under high pressure, the bonds between carbon materials are broken to form bioactive GQDs. The electrical-optical properties of the particles can be adjusted by using different precursors and through temperature optimization. Therefore, the hydrothermal method is low-cost and non-toxic (Figure 3. Hydrothermal methods also produce particles with a diameter of 10 nm, more prominent than other dot production methods, such as the electrochemical method [120,121,122,123,124,125]. Lianget al. easily obtained highly fluorescent quantum dots from gelatin by using hydrothermal method [126]. Liu and colleagues developed a simple, green, and inexpensive way to prepare fluorescent quantum dots using hydrothermal processing and a type of radish. They investigated the application of prepared carbon quantum dots in cell imaging and the detection of iron ions [127]. Hong Miao et al. presented a green method for synthesizing carbon quantum dots with a high quantum yield of about 14% from tomato paste and investigated the use of synthesized quantum dots in determining the type of antigen [128].

#### 2.5.2. Microwave Method

The synthesis of nanoparticles via microwave-facilitated methods has many advantages compared to other methods. Compared to the hydrothermal method, this method has the advantage of its higher speed and lower synthesis temperature. Some of the benefits of microwave facilitated reactions are: (1) Deficient level of impurities in the products, (2) a very high product efficiency, (3) the easy control of temperature and pressure, (4) the easy control of product properties (such as size), (5) the environmental friendliness of the method, (6) the very high security of the method, (7) its reproducibility, and (8) selective heating (i.e., the reduction of energy costs) [129,130,131]. Ayele et al. have proposed a green method for the mass production of CdSe quantum dots by using microwaves [132]. An easy, light, and fast method for the synthesis of nitrogen-doped GQDs using microwave prepared cedar tree root without surface modification has been proposed by Shawing Sheng et al. [133]. The resulting quantum dots were used to identify Hg2 + ions selectively. Synthesized GQDs can also be used as multicolor fluorescents in bioimaging. The purification of dialysis membranes for 7 h and the preparation of GQDs via microwave irradiation for 10 min using raw materials such as aspartic acid (Asp), NH4HCO3, and solvent (DI water) were investigated by Zhang et al. The results showed that the prepared GQDs had a strong blue fluorescence and a QY of 14% (Figure 4) [134].

#### 2.5.3. Electrochemical Oxidation Methods

Electrochemical methods make it possible to adjust nanostructures by controlling the current–voltage ratio. For example, applying a regulated voltage to a bulk of carbon precursors leads to the electrochemical corrosion reactions of carbon reactants and the production of carbon nanostructures. This method does not require high temperatures; it can be done quickly on a large scale with aqueous or non-aqueous solvents. This method is one of the fastest ways of producing graphene sheets (Figure 5) [103,104,105,135,136]. For example, Paulo et al. prepared GQDs using electrochemical and corrosion methods [137]. Wong et al. proposed a bottom-up approach for synthesizing nitrogen-doped GQDs using the electrochemical process. This method is green and simple and is suitable for large-scale synthesis with an interest rate of more than 95%. The quantum gain of the particles is 0.71. Also, the toxicity of quantum dots synthesized by this method is low [138].

#### 2.5.4. Ultrasonic Method

In recent years, many methods for making photoactive materials for use in solar cells and photocatalysts have been proposed. However, the benefits of preparing these materials in unusual ways are exciting from the point of view of green chemistry. The idea of producing highly efficient photocatalysts through the use of ultrasound is fascinating and essential in terms of science and technology and has excellent potential for producing photocatalysts in the near future. Ultrasonic synthesis is a promising method that allows for control over size, morphology, nanostructure, and catalytic properties. Compared to conventional energy sources, the supersonic method provides abnormal reaction conditions in the liquid phase due to the phenomenon of cavitation (very high temperatures and pressures for brief periods in liquids) [139,140,141]. Zhu et al. synthesized high-quantum GQDs from only graphene oxide and potassium permanganate in a one-step method using ultrasonic irradiation and used them as a test for the detection of alkaline phosphate (ALP) [142]. Oza et al. prepared GQD quickly and greenly using the precursor chemical acoustic method and a lemon [143]. The resulting quantum dots were purified using the sucrose gradient density centrifugation method, which separated the water-soluble, single-spray, photoluminescent, highly stable optical, and chemical quantum dots. Furthermore, quantum dots synthesized in this way are not toxic and can be used as optical imaging tools. The formation of tens of thousands of small bubbles in the liquid and the loss of carbon–carbon bonds are generated by mechanical force and eventually results in GQD cutting, an issue which is affected by ultrasound waves (Figure 6).

#### 2.5.5. Biosynthesis Method

The biosynthesis method is performed in mild conditions, and the bioactive GQDs synthesized using this method have inherent biocompatibility and bio-stability without the need for encapsulation operations. Therefore, these methods provide a green way to prepare biocompatible, bioactive GQDs. As with chemical processes, biosynthesized quantum dot emissions can be controlled by their size, the instantaneous excitation of bioactive GQDs of different sizes by a single light source, and by a broad spectrum window that can be adjusted by the constructive component [107,108,144,145].

### 2.6. Green Reduction of GQD

Advances in nanoscience and nanotechnology have led to versatile applications and the discovery of different new nanomaterials, the properties of which depend on their shape and size. QD is known as one of the nanomaterials that has been introduced due to its variation in bandwidth size. Since there are numerous challenges involved in disposing and recycling of QDs, it can be said that they are very dangerous to process, specifically ordinary QDs such as PbS and CdSe [146]. Bioactive GQDs, an example of organic quantum dots, are not as limited as conventional QDs. Bioactive GQDs, known as nanomaterials, have advantages such as adjustable diffusion, excellent biocompatibility, resistance to light bleaching, the abundance of raw materials in nature, and low toxicity and costs [147]. Among their potential applications are their use as light-emitting diode materials [148], detection probes [149], and optical bioimaging probes. An effective way to regulate the electrical and optical properties of GQDs is to dope GQDs with heteroatoms (such as B, N, S, and F) [100]. In GQDs, the presence of N can be effective for the homogeneous distribution of metal nanoparticles. Nevertheless, N-GQDs, in addition to being generated in situ for catalytic proposals, can support metal nanoparticles such as Pd (0). N-GQDs were expected to be an adequate support for Pd (0) in the catalytic reduction of nitro compounds due to the efficient activity of graphene as a support in the green reduction reaction [150].

## 3. GQD Based Polymer Composite

In graphene-based polymer composites, the superior properties of graphene compared to polymers are reflected. Compared to neat polymer, graphene-based polymer composites show gas barrier, electrical, superior mechanical, flame retardant, and thermal properties [41,42,151,152,153,154]. Graphene nanofibers are used as a two-dimensional model to arrange polymers, increasing the polymers’ solubility [155]. Although carbon nanotubes (CNTs) exhibit mechanical properties comparable to graphene, graphene is a better nanofiller in certain respects, such as in terms of thermal and electrical conductivity [156,157,158,159,160,161]. Interfacial bonding between graphene layers and the polymer matrix and the distribution of graphene layers in the polymer matrix affect nanocomposites’ physicochemical properties. Pure graphene does not have good compatibility with organic polymers and does not form homogeneous composites. Since the use of graphene oxide (GO) sheets as nanofillers has attracted much attention in the context of polymer nanocomposites, it can be said that in contrast to graphene, GO is preferable to organic polymers [162,163,164,165]. Graphene oxide is an insulating material, so it cannot synthesize conductive nanocomposites without further resuscitation. In polymer, graphene, and solvent, factors such as polarity, molecular weight, hydrophobicity, and reactive groups play an essential role in when it comes to preparation methods [166]. For advanced hybrid nanomaterials and applications, polymer dots have been designed and used, specifically necessary and unique quantum dots. Therefore, polymer dots can be obtained by using conjugated and non-conjugated polymers [167,168]. Both QD polymer encapsulation, without changing the native surface ligand layer [169], and ligand exchange, used to replace the main hydrophobic ligands on the GQD surface with polymer molecules [170], are methods of making GQD-polymer nanocomposites. Biological imaging, fluorescence imaging, and bioassay from GQD-based nanocomposites are used [171,172,173]. The foundation of potentially new and innovative GQD composites [174,175,176,177] has been established through previous research on graphene-reinforced composites [178,179,180] and GQD synthesis [20,181,182,183]. One of the newest forms of graphene is GQDs. Electronic devices [166,184], energy storage [185,186], fuel cells [187], and biomedical applications [188,189] are among the wide range of applications of GQDs-polymer nanocomposites.

## 4. Polymerization

Polymerization is a chemical reaction in which small and simple molecules, or monomers, bond together to form a large molecule with a molecular mass several times that of the original molecule [190,191,192]. Bulk, emulsion, suspension, and surface polymerization are polymerization methods used to prepare microparticles. One or more monomers are heated in a bulk polymerization technique to induce polymerization in the presence of a catalyst. During the polymerization process, the drug may form. Also, monomers can be prepared or fragmented to convert the obtained polymer into a microsphere [193]. The polymerization process occurs at lower temperatures, and the polymerization of suspension is similar to bulk polymerization [194].

### 4.1. Methods for Polymerization

#### 4.1.1. Suspension Polymerization

To obtain a hybrid of stable QDs of polymer particles, considering the confinement of QDs during a polymerization reaction, one of the investigated methods is the polymerization of suspension [195,196,197]. Hexadecylamine CdSe QDs were synthesized by O’Brien et al. The surface polymerizable QDs were then obtained via ligand exchange. As a result, the QDs were encapsulated in the synthesized polymer particles [198]. The inclusion of QDs in the polymer particles during the polymerization process of suspension results in reasonable control with respect to the position of the quantum dots or their final fluorescence properties.

#### 4.1.2. Emulsion Polymerization

The enclosure of QDs in colloidal polymer particles is achieved through emulsion polymerization [199,200,201]. The coverage of CdSe quantum dots with tridactyl phosphine oxide and their dispersion in toluene were investigated by Yang and Zhan. The results show good control over the confinement of quantum dots in polymer particles to prevent the accumulation of nanocrystals and fluorescence microscope images [202]. It is not easy to completely enclose QDs in polymer particles using emulsion polymerization unless polymerization starts from the surface of QDs after their surface has been modified to combine polymerizable groups [197].

#### 4.1.3. Mini-Emulsion Polymerization

Mini-emulsion polymerization is one of the simplest and most common methods for encapsulating QDs in hydrophobic sub-micron polymer particles. A polymerization monomer produces an aqueous dispersion of nanoparticles where the surface of the coated QDs is compatible [203,204,205,206,207]. The creation of polymer nanoparticles containing QDs is achieved through the polymerization of a dispersion nanodroplet. The monomer system, the type of initiator, the surfactant concentration, and the surface modification of the mineral nanoparticles are among the factors on which the successful encapsulation of the mineral nanoparticles in the polymer particles by mini-emulsion polymerization depends [208].

### 4.2. Physical Blending

The physical blending approach makes it easy to prepare GQD/polymer nanocomposite materials. Some physical blending methods include mixing tiny amounts of GQD with a polymer matrix, blending the solution, and processing the melt. Hydrogen bonding, electrostatic interactions, or π-π interactions between polymers and GQDs are among the non-covalent interactions that lead to nanocomposite formation. When water molecules stick together, an “electrostatic” bond is formed between an oxygen atom from one molecule (which has a negatively charged particle) and a hydrogen atom from another molecule (which has a positively charged particle), and this is a hydrogen bond. The distribution of attractive driving force interactions when using this method to prepare GQD/polymer composites led to the creation of suitable quality nanocomposites with good GQDs [209,210,211,212,213,214].

#### Infusion of GQD into Polymer

Injecting a bandgap through modifications that introduce quantum confinement and edge effects makes graphene nanoparticles such as GQDs and graphene nanoribbons [215]. A subset of graphene-based nanomaterials consisting of several graphene layers stacked on top of each other are called GQDs [216]. Glucose [216], carbon fibers [217], carbon nanotubes (CNTs) [218], graphite [219], and coal are among the various materials used in the synthesis of GQDs. Gobi et al. investigated the injection of graphene quantum dots to create stronger, harder, and brighter polymer composites. When comparing polymer resins with and without GQD, uniform loading with weight percentages of up to 10%, an 18% change in the maximum tensile strain, and a 2.6-fold increase in toughness, etc., are the results of GQD injection into an epoxy polymer matrix, reflecting the excellent optical properties of the composite formulation [182].

### 4.3. Chemical Grafting

Esterification [220], etherification epoxidation [221], methacrylation/acrylation [222], and acylation are among the abundant functional groups on the GQD surface that allow for chemical modifications to form covalent bonds with polymers due to the covalent bonds between polymer chains and GQDs. The chemical grafting of GQDs to the polymer matrix is superior to the physical blending approach in terms of promoting mechanical strength and maintaining properties over time [223,224,225,226,227].

### 4.4. In-Situ Growth

Since the secondary intermolecular interaction between GQDs and the polymer matrix is relatively weak and unstable, it can be said that the physical blending method is a practical approach that is used to fabricate GQD/polymer nanocomposites. In many cases, chemical grafting methods use organic reagents, and solvents are toxic due to their shortcomings in terms of complex reaction processes and multiple preparation steps. The physical blending method can also anchor GQDs firmly to the polymer matrix due to the formation of covalent bonds. The ensuring of a better bond strength is possible via the in-situ growth of GQDs in the polymer matrix because it is an easy and green process. Also, the bond between GQD and polymer includes both chemical and physical interactions. An in-situ growth strategy can be obtained by applying one-pot thermal treatment containing hydrothermal, pyrolysis, or low temperature heating on a mixture containing GQD precursors and polymer solution [228,229,230,231,232].

## 5. Biomedical Applications

### 5.1. Drug Delivery

Bioactive GQD based polymer composites can bind to various biological materials through π-π and electrostatic interactions, so the loading of most drugs can be done quickly. On the other hand, bioactive GQD based polymer composites have good membrane permeability and biocompatibility, which can improve the drug efficiency of the loaded drug and increase effectiveness in the face of drug-resistant cells [233,234]. Table 2 shows the application of bioactive GQDs in the field of drug delivery. Extensive research has been done on using bioactive GQD based polymer composites for in vivo drug delivery. Absorption by the reticuloendothelial system and renal clearance is reduced due to the average size of bioactive GQDs. The efficiency of bioactive GQDs can also be attributed to their high delivery, which increases blood circulation time [235]. Sam et al. reported the high anticancer activity of GQDs containing curcumin in both in vivo and laboratory conditions. They combined three types of curcumin-graphene composites, including GOs, DGOs, and GQDs, to form the hydrophobic cancer drug curcumin. Interactions between curcumin and the oxygen-containing functional groups DGOs, GOs, and GQDs, played a crucial role in the loading capacity of curcumin. Since the number of oxygen-containing functional groups on the surface of the graphene derivatives was pH-dependent, the effective drug loading and release behavior was pH-dependent. According to the results, the prepared GQD-curcumin composites contained the highest amounts of curcumin (40,800 mg/g). They had the best anti-cancer activity compared to other composites containing the same dose of curcumin. There is no fluorescence in the case of curcumin and its GQD composite. It was only after the release of curcumin from the composites of GQDs that the GQDs provided the remaining fluorescence signal. As a result, GQDs simultaneously act as probes for tumor imaging along with drug release [236]. Silica-coated bipolar quantum dots were investigated by Akbarzadeh et al. They reported that the synthesis of intelligent nano-aptamer was targeted by QDs coated with porous meso silica with a bimodal imaging capacity. Their results showed that the targeted hybrid system that was prepared based on their in-vivo test has MR capability and fluorescent imaging [237]. Iannazzoa et al. examined the use of graphene QDs in cancer treatment and drug delivery. Based on the results, the prepared GQDs have a remarkable ability to deliver the drug to cancer cells and are biocompatible. Similarly, it is possible to combine targeted ligand drugs in nanomaterials via a multifaceted combination. As a result, conventional chemotherapy, toxicity and side efficacy were minimized (Figure 7) [238].

### 5.2. Gene Delivery

Gene therapy is one of the new and promising ways to treat various diseases caused by genetic disorders such as cystic tissue corruption, Parkinson’s disease, and cancer [244]. Figure 8 shows the treatment of cancer cells with gene therapy based on bioactive GQDs. Gene therapy for treating a wide range of disease conditions such as rare hereditary and single-gene disorders has also attracted much attention. For this purpose, nucleic acids must be delivered to human target cells and expressed. Gene therapy offers a new treatment method by offering DNA or RNA instead of chemotherapy drugs to inhibit cell pathways and induce tumor cell death. This treatment method can also remove a malfunctioning gene, replace the mutated gene, or offer new gene products to fight the disease [245,246,247]. Successful gene therapy requires a gene carrier that can protect DNA from nucleoside degradation and, in addition, facilitate high-yield DNA cell uptake [248]. The primary challenge facing the development of gene therapy is the lack of safe and highly efficient carriers [249]. Among the effective non-toxic carriers in gene therapy are none-viral gene delivery systems. Such systems ensure that nucleic acid payloads to cytosols and cell nuclei bypass biological barriers. Viral vectors are one of the factors influencing gene delivery due to their natural ability to invade and deliver genetic material. It is also possible to keep them safe for clinical use by blocking the immunogenicity of viral vectors [250].

Bioactive GQD-based polymer composites demonstrate the characteristics of an effective gene carrier. Therefore, many efforts have been made to develop bioactive GQD-based polymer composites as non-viral vectors in gene therapy. The development of bioactive GQD-based gene nanocarriers is possible by forming a set of bioactive GQD-based polymer composites, plasmid DNA (pDNA), and chimeric peptides. Observing the cellular uptake of GQD-peptide-pDNA complexes is possible concerning PL bioactive GQD-based polymer composites through a confocal microscope [251]. Based on previous studies, the use of bioactive GQD-based polymer composites as transmission vectors can lead to more efficient in-vitro and in-vivo gene therapy applications. The application of bioactive GQD-based polymer composite on other non-viral vectors seems to be less common than using bioactive GQD-based polymer composites as drug delivery agents. Due to the sp2 hybridized structure and π-stacking in bioactive GQDs-based polymer composite systems, the possibility of drug loading in bioactive GQD-based polymer composite systems is higher than in other nanoparticle-based drug delivery systems [252,253,254,255,256].

### 5.3. Photothermal Therapy (PTT)

The therapeutic window refers to the aromatic structure that provides graphene with strong light absorption in the NIR region (700–900 nm) [257,258]. The main treatments of cancer are radiology and chemotherapy, which have significant side effects. Recently, PTT has been considered due to its high efficiency in destroying malignant tumors as well as the minor amount of damage it inflicts on adjacent healthy tissues (Figure 9) [259]. In the photothermal therapy method, electromagnetic wave radiation (with wavelengths in the infrared region) and absorbent nanoparticles at these wavelengths, which have a high efficiency of light to heat, are used. Thus, by converting light energy into heat and increasing the temperature to between 42–46 °C, cell membranes and proteins are destroyed, leading to the death of cancer cells [260,261]. Many nanoparticles are identified as external agents after entering the body due to their hydrophobicity and cause poisoning by accumulating in one part of the body. Therefore, the surface of nanoparticles is coated with ligands to increase biocompatibility and reduce toxicity [262]. Nanoparticles used in cancer treatment via PTT include gold nanoparticles, carbon nanotubes, and graphene. The efficiency of light production to heat in graphene and its biocompatibility are both higher when compared to gold nanoparticles and carbon nanotubes.

On the other hand, due to the increased ability of graphene, the use of these nanoparticles has received much attention when it comes to cancer drugs and the simultaneous treatments of PTT and drug delivery. By placing graphene on the curved surface of the nanoparticle, the interaction of light with graphene increases, increasing the efficiency of PTT [263]. The features of this therapeutic method are non-invasive radiation, penetration into the skin, and its harmlessness. In recent years, graphene-based PTT nano-hybrids have been discovered to inhibit tumors in vivo. After intravenous injection, GO complexes begin to accumulate in the solid tumor severely and passively, due to the effect of EPR. The effective death of cancer cells occurs when GO complexes produce enormous heat after irradiation. Recently, PTT in vivo using GO and GO-complexes was reported to remove 100% of tumors [264]. The successful treatment of cancer cells was demonstrated by using nitrogen and boron dual doped bioactive GQD-based PTT in the near-infrared II region [265]. The effectiveness of such treatment is achieved by merging a drug in bioactive GQD-based polymer composites with PTT. The bioactive GQDs-gated hollow mesoporous carbon nanoplatform loaded with doxorubicin achieves this effect simultaneously with the controlled release of near-infrared drug delivery [266]. This combination of PTT and PDT is exciting. Thus, a multifunctional bioactive GQD is formed to treat cancer when it produces 808 nm ROS laser radiation and heats up simultaneously [267].

### 5.4. Photodynamic Therapy (PDT)

PDT activates a light-sensing compound by using visible light to generate samples of cytotoxic oxygen and free radicals that selectively destroy rapidly growing cells [268]. PDT is widely used to treat acne, psoriasis, and cancer using a light source and a light-sensitizing agent. Light sensitizing factors such as semiconductor quantum dots have received much attention in the context of PDT. The killing of target cells occurs via a PDT mechanism involving ROS production and oxidative stress. ROS production in cancer cells and tumors is mediated by a light-sensitizing agent such as a bioactive GQD-based polymer composite (Figure 10). The factors that make bioactive GQD-based polymer composites a good light sensitizing agent include stability in various pHs and light, their biological corrosion resistance, and their biocompatibility. To achieve PDT and the simultaneous imaging of cancer cells, bioactive GQD-based polymer composites can be used. The prevention of photobleaching and the production of a high quantum efficiency from a single oxygen molecule are among the benefits of bioactive GQD-based polymer composites. In PDT, bioactive GQD-based polymer composites are more practical than any other factor. The induction of cancer cell apoptosis and autophagy is performed by bioactive GQD-based polymer composites through oxidative stress.

Studies have shown that cancer treatment using bioactive GQD-based polymer composites is more effective than photodynamic therapy using HeLa cells [269,270,271,272]. Chemotherapy and photodynamic therapy guarantee a synergistic effect in the treatment of cancer. With GQDs, the targeted PDT of mitochondria is also performed [273]. Reports on the PDT of two photons were performed using a Bengal rose light sensitizer with nitrogen-doped GQDs [274]. A synergistic effect on tumor cell apoptosis was demonstrated simultaneously with drug delivery and light irradiation by silver nanoparticles of PEGylated GQDs loaded with doxorubicin (DOX) [275].

### 5.5. Bioimaging Application

Bioimaging is a method used in both clinical and research settings. Using different parts of the electromagnetic spectrum means that the bio-distribution of therapies separately and accurately and the observation of biological processes such as targeted delivery provides cellular uptake [22,234,276]. The application of QDs in cell imaging is multicolored and sensitive due to a significant increase in their synthesis, surface composition, and chemistry [277]. The extensive application of bioactive GQD-based polymer composite in biological imaging are stimulated by broad absorption with narrow emission spectra, strong quantum confinement, and relatively high quantum efficiencies with high molar extinction coefficients [278,279,280]. Bioactive GQD-based polymer composites are promising candidates for imaging deeper tissue samples because they have the property of emitting near-infrared reflection [281]. Since the detection of metastasis and the recurrence of cancer, and the early detection of tumors by sensitive imaging, are possible, it can be said that the role of imaging in the diagnosis of cancer is of particular importance. Figure 11 illustrates the diagnosis of cancer using bioimaging based on bioactive GQDs. As required with other nanoparticle platforms, the intrinsic PLs of GQDs allow them to be used as optical probes in fluorescence imaging without the further combination of fluorescent dyes. Recently, the capabilities of bioactive GQD-based polymer composites in MR imaging and NIR fluorescence imaging have led to the development of bioactive GQD-based polymer composite in bioimaging. Among the unique features of bioactive GQD-based polymer composite, we can mention their ideal imaging probes, with applications in different bioimaging methods, and their excellent biocompatibility [256]. In Sheng et al., the doping of quantum dots with nitrogen (N-GQDs) was performed using a hydrothermal process. In this study, raw materials such as citric acid, PVP K90, and glutamate were used, and the quantum efficiency was 64.2%. The results showed that a new fluorescence probe of nitrogen-doped GQDs could be used to detect chromium (VI) and mark MCF-7 cells [282].

### 5.6. Tissue Engineering

Tissue engineering is a method in which cells are taken from a patient, and after culturing and increasing their number, they are accepted in a scaffold. Appropriate chemical, biological, mechanical, and electrical stimuli are applied, and new tissue is formed in a short time [283]. The main components of tissue engineering are the cell, the scaffold, and the growth factor [284]. More recently, bioactive GQDs have been tested for stem cell-based tissue engineering and reconstructive medical applications. Many efforts have been made to differentiate stem cells into different cells by performing other techniques [285]. Encouraging specific stem cell differentiation under certain conditions is possible using bioactive GQD-based polymer composites. The important role of bioactive GQDs in bone differentiation was investigated by Qiu et al. [286]. The critical role of bioactive GQDs in damaged bone formation in tissue engineering is shown in Figure 12. Specifically, bioactive GQDs were found to stimulate the primary activation of ontogenesis. In medicine, bioactive GQD-based polymer composites are considered valuable in terms of regeneration because these particles have excellent mechanical properties, differentiation powers, and low toxicity. In addition, they can increase the abundance of calcium [16]. In tissue engineering, the improvement of the mechanical properties of scaffolds in which orthopedic implants can be placed is performed by bioactive GQDs, which act as reinforcing agents [287]. In addition to bone, bioactive GQDs are used for nerve and cartilage tissues [16].

## 6. Cytotoxicity and Biocompatibility

The size and shape of GQDs, the doping of non-metallic ions, and the modification of graphene surfaces and edges by photoluminescence (PL), one of the most prominent properties of graphene, are all easily adjustable [288,289]. A Large surface area, easy surface modification, high photostability, low cytotoxicity, and excellent biocompatibility, besides adjustable PL, are all features of GQDs [290,291]. Compared to conventional semiconductor QDs and other carbon-based luminescent nanomaterials, photoluminescent GQDs are more durable. GQDs are used in biomedical applications due to their adjustable PL, excellent biocompatibility, low cytotoxicity, high biocompatibility, and ease of functionalization, and, as imaging and labeling luminescent agents, are attractive in the context of tracking molecular targets in living cells or organs [292].

## 7. Conclusions and Perspective

In the last few decades, bioactive GQDs have attracted much attention due to their properties and applications in various environmental and health fields. There are problems that need to be addressed in the context of their practical biological applications, such the preparation of bioactive GQDs, their size, reproducibility, and low quantum efficiency. The size of bioactive GQDs is responsible for their fluorescence properties. The low toxicity of bioactive GQDs also makes them suitable for use in many in vivo applications. Therefore, it is possible to expand their potential applications in various fields by overcoming the problem of their low quantum efficiency by preparing bioactive GQD nanocomposites with surface factorization and band gap engineering. In this study, bioactive GQDs were introduced, and their synthesis methods were explained. The various biomedical applications of bioactive GQDs, such as bioimaging (in vivo and in vitro), drug delivery, gene delivery, photothermal therapy, photodynamic therapy, and tissue engineering, were discussed. The effortless combination of many drugs and ligands is provided through the large surface and functional groups of bioactive GQDs. Taking advantage of this, bioactive GQDs can be used as a nanocarrier for targeted drug delivery. Also, the photoluminescence of bioactive GQDs is used to develop bioimaging techniques to identify various biomolecules that offer a wide range of new strategies for disease diagnosis. They can be used to image cancer cells and to track drug delivery to cells and tissues. Bioactive GQDs present low levels of toxicity in the human body while having adjustable optical properties and fluorescence propagation in the NIR region of the light spectrum. Researchers aim to develop bioactive GQDs for biomedical applications and face a key challenge in the form of achieving high-quality products. The small-scale production of bioactive GQDs with large size distributions is possible through existing synthesis methods. Therefore, it is necessary to find new methods to achieve high performance and easy purification methods that do not require the removal of raw materials. Recent research advances in polymer composites based on bioactive GQDs focusing on their synthesis and biomedical applications, including bioimaging (in vivo and in vitro), drug delivery, gene delivery, light therapy, photodynamic therapy, and tissue engineering, were summarized in this review. Finally, we conclude that there is a promising future for further developing GQD-based polymer composites for many unresolved therapeutic barriers.

## Figures and Tables

**Figure 1 polymers-14-00617-f001:**
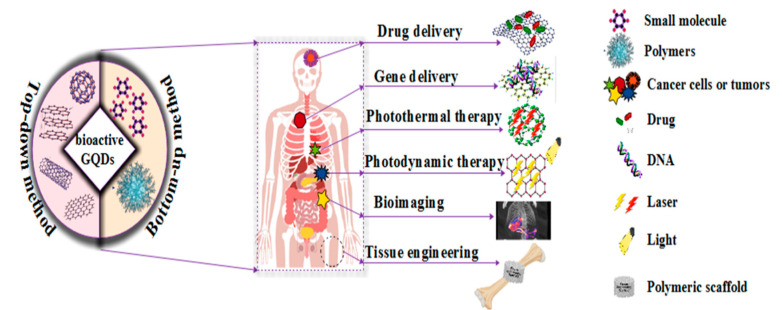
The applications of bioactive GQD-based polymer composites in biomedicine.

**Figure 2 polymers-14-00617-f002:**
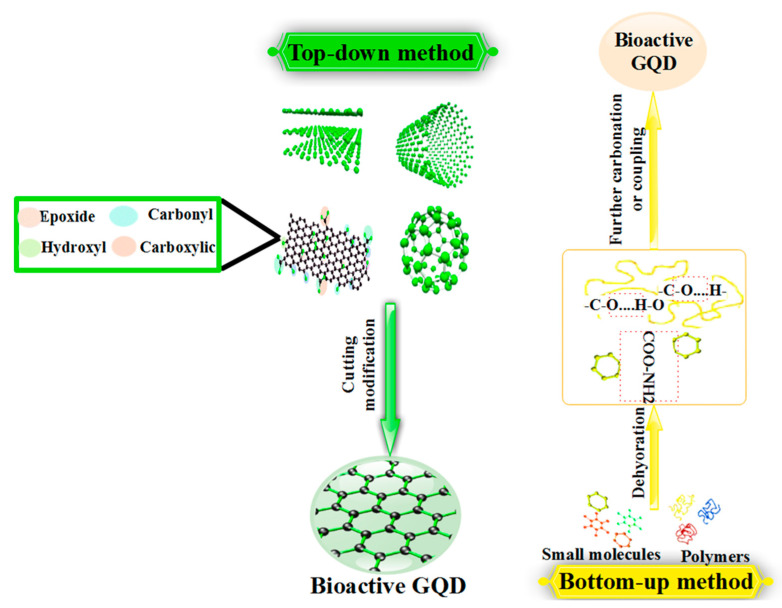
The top-down and bottom-up methods for the synthesis of bioactive GQDs.

**Figure 3 polymers-14-00617-f003:**
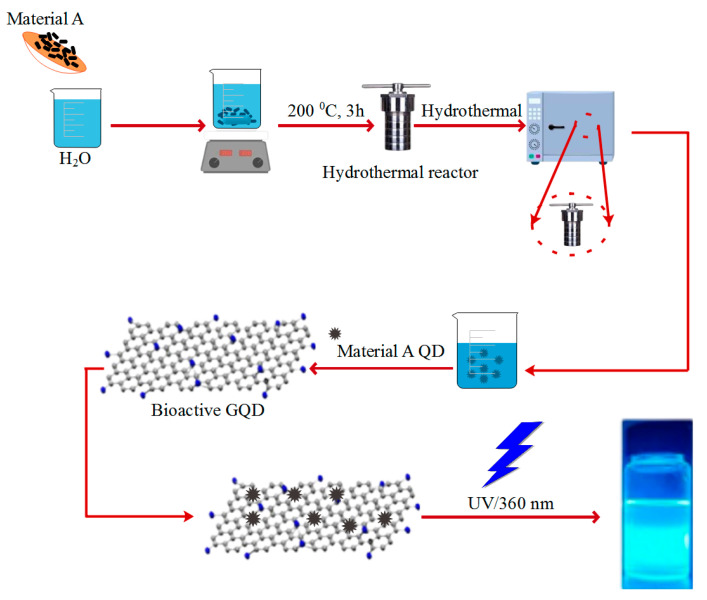
Synthesis of photoluminescence GQDs by using hydrothermal method from material A.

**Figure 4 polymers-14-00617-f004:**
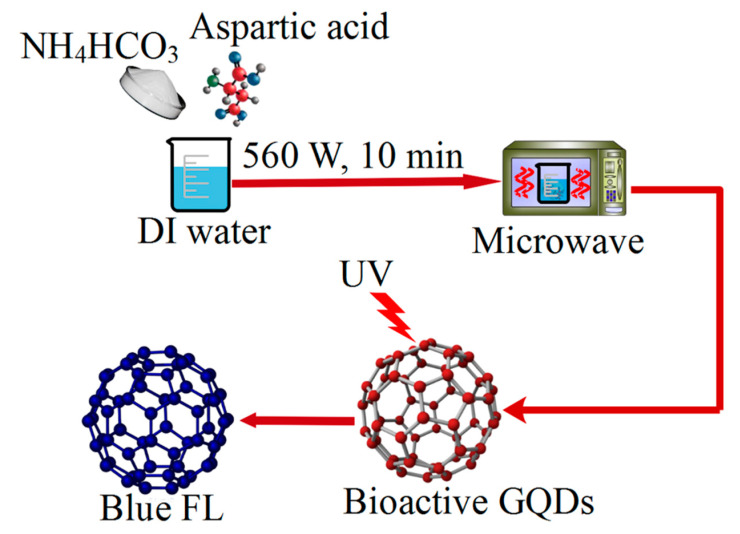
The purification of GQDs using microwave irradiation.

**Figure 5 polymers-14-00617-f005:**
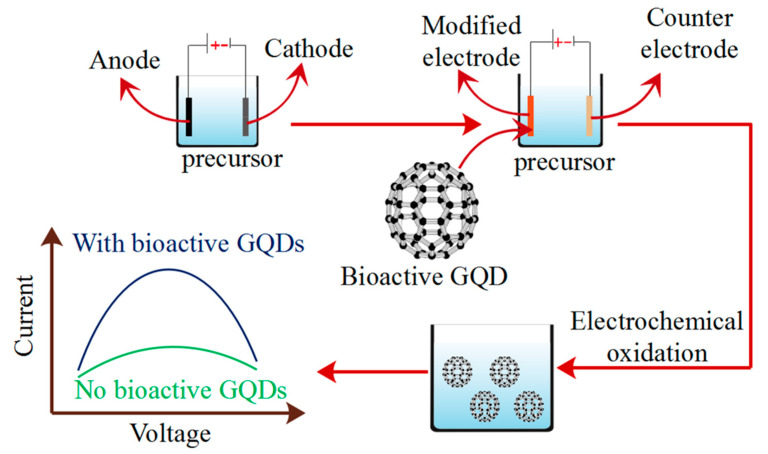
The synthesis method of electrochemical oxidation.

**Figure 6 polymers-14-00617-f006:**
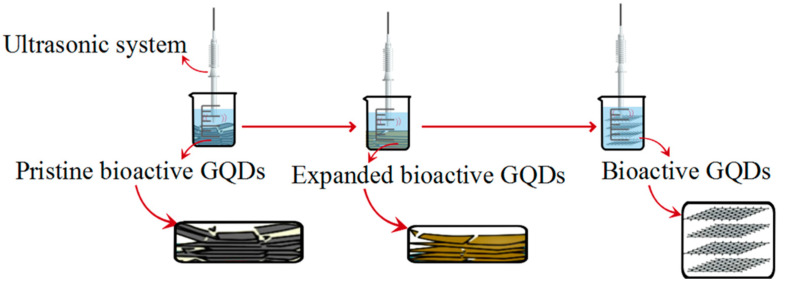
The ultrasonic synthesis method.

**Figure 7 polymers-14-00617-f007:**
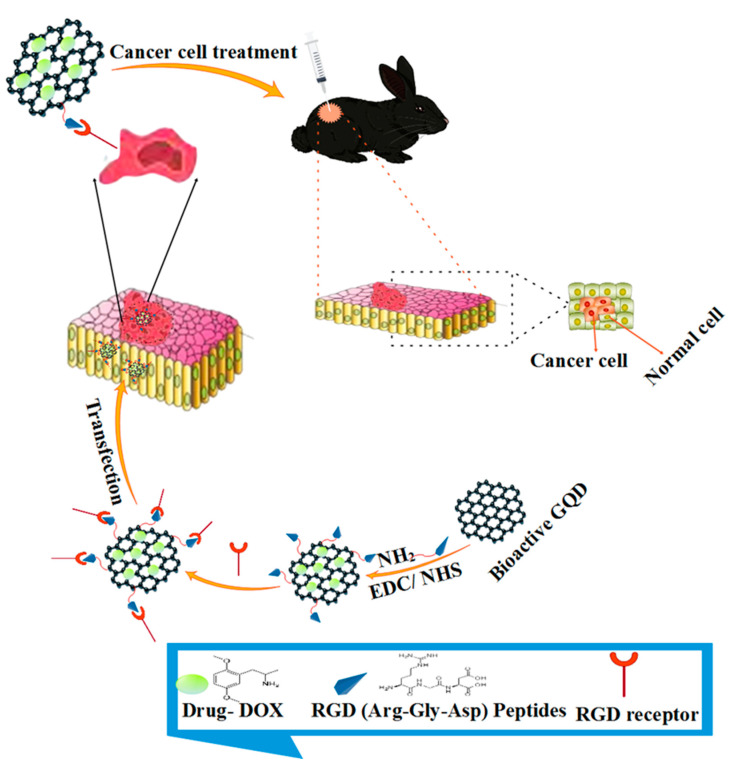
Treatment of cancer cells with drug delivery method based on bioactive GQDs.

**Figure 8 polymers-14-00617-f008:**
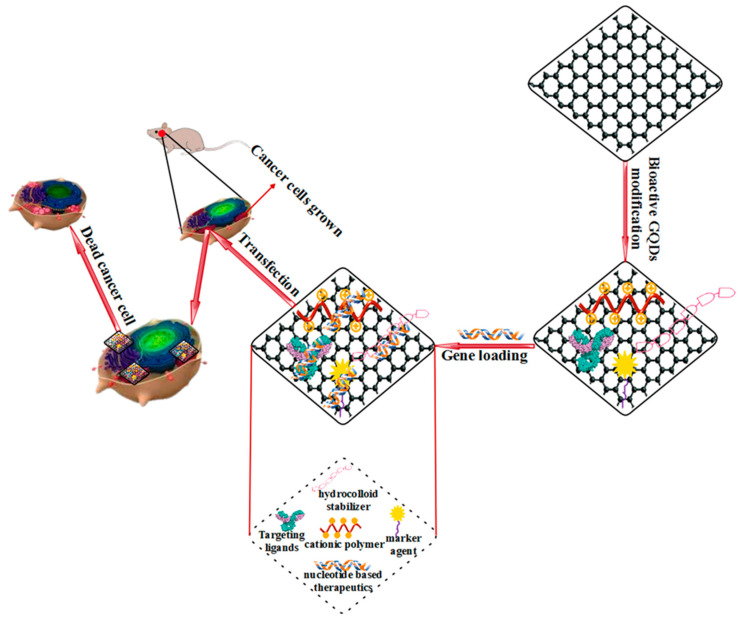
Treatment of cancer cells with gene therapy based on bioactive GQDs.

**Figure 9 polymers-14-00617-f009:**
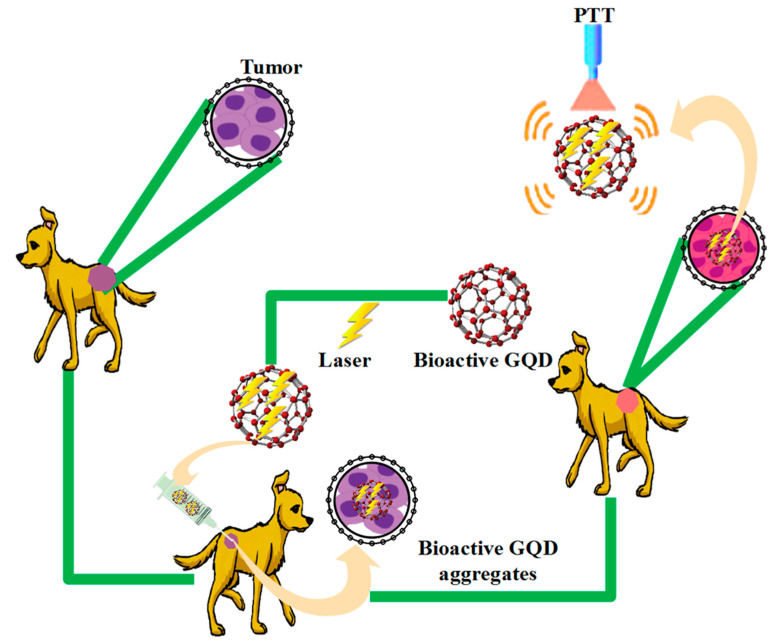
Tumor treatment with the PTT method based on bioactive GQDs.

**Figure 10 polymers-14-00617-f010:**
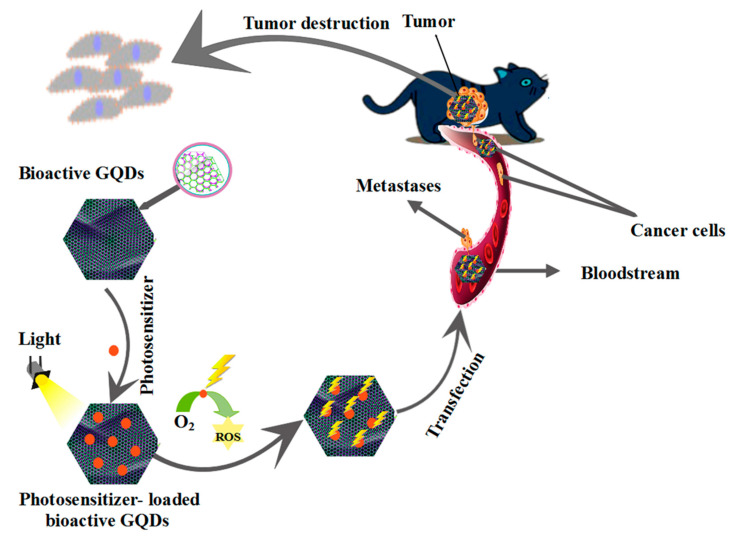
Tumor destruction using the photodynamic method based on bioactive GQDs.

**Figure 11 polymers-14-00617-f011:**
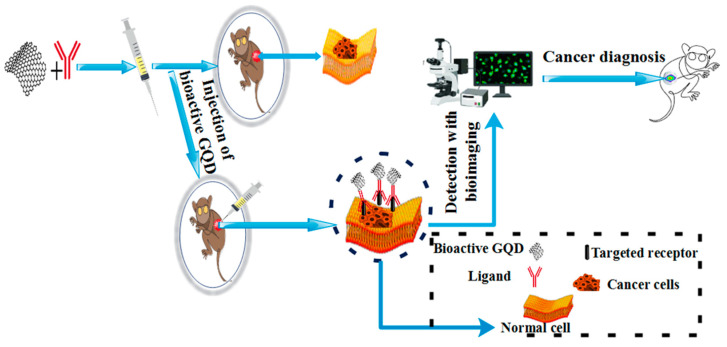
Diagnosis of cancer using bioimaging based on bioactive GQD.

**Figure 12 polymers-14-00617-f012:**
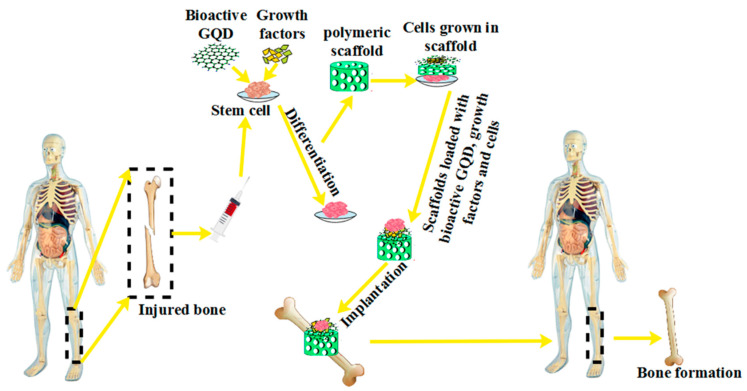
The important role of bioactive GQDs in damaged bone formation in tissue engineering.

**Table 1 polymers-14-00617-t001:** Different methods for synthesizing bioactive GQDs are used in biomedical applications.

Synthesis Methods	Application	Size	Ref.
**Bottom-Up Method**			
GQD-RhB-silka	Diagnosis	3–20 nm	[109]
mango leaf extract—mGQDs	NIR-responsive fluorescence bioimaging	2–8 nm	[110]
PEGylated GQD	Fluorescence imaging of tumors	2.75 nm	[111]
GQD-PEI	Gene transfection	3–4 nm	[112]
GQDs	Drug delivery and bioimaging	∼12 nm	[113]
MSN-SS-CD-DOX	Targeted and controlled drugs	2.7 nm	[114]
**Top-Down Method**			
GQD	Diagnosis	5 nm	[115]
durian extract—GQDs	Bioimaging	2–6 nm	[116]
NP-GQD	Cysteine detection	10–30 nm	[117]
GQD-PEG-AG	Radiotherapy	3–4 nm	[118]
lignin—GQDs	Bioimaging	2–6 nm	[119]

**Table 2 polymers-14-00617-t002:** The application of bioactive GQDs in the field of drug delivery.

	Composite	Application	Ref.
1	GQD/DOX	In vitro: A549 cells	[239]
2	SiRNA/GQD/DOX	Therapy of A549 cancer cells	[240]
3	GQD/DOX	In vitro: HeLa, A549, and HEK293A cells	[241]
4	CMC/GQDDOX	Therapy of K562Leukemia cells	[242]
5	GQD/DOX	In vivo: BALB/c mice	[234]
6	GQD/CDDP	Breast cancer cells	[243]

## Data Availability

All data generated or analyzed during this study are included in this published article.

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
