# Peer review of "Bioactive Graphene Quantum Dots Based Polymer Composite for Biomedical Applications"

_polymers, 2022, doi:10.3390/polym14030617_

Round 1
Reviewer 1 Report
In this work, bioactive graphene/quantum dots polymer composites for biomedical applications are reviewed.
I would like to address the following comments to the Authors:
- The originality and novelty of this review are unclear. What makes this review stand out from other recent reviews on the topic? This must be highlighted.
- The aim of the study in the last paragraph of the Introduction indicates that the focus of this paper is on the synthesis methods of bioactive graphene quantum dot-based polymer composites. However, only 1.5 pages of the article are devoted to the synthesis methods, while as many as 7 pages are devoted to the application areas. The aim should therefore be reworded.
- The description of polymer nanocomposites in the last paragraph on page 2 is not entirely accurate and clear. The definition of polymer nanocomposites (page 2, lines 81-82) does not correlate fully with the EU adopted definition of a nanomaterial (2011/696/EU). In the next sentence, polymer nanocomposites are compared with conventional nanocomposites (page 2, lines 82-84). What does "conventional nanocomposites" mean in this sentence?
- Chapter 2 cites only one literature source from the last 5 years, despite the fact that many articles and even reviews have been published on this topic. I suggest that this chapter be expanded and supplemented with more recent literature.
- The text of the Conclusions is very poor and general. Recent advances and perspectives are not specified. The meaning of the first two sentences is the same.
- There are many spelling and grammar errors in the text.
Author Response
Manuscript Ref. No: polymers-1561996
Bioactive Graphene quantum dots Based Polymer Composite for Biomedical Applications
Dear Editor-in-Chief Journal of Polymers,
Thank you for your letter and the opportunity to revise our manuscript ‘Bioactive Graphene quantum dots Based Polymer Composite for Biomedical Applications’ in the journal Computational and Mathematical Methods in Medicine. We appreciate you and the reviewer for the time and effort dedicated to providing thoughtful comments and constructive suggestions.
Below, the reviewer’s comments and cognate responses are written one by one, indicating exactly how we addressed each concern or problem and describing the changes we have made. The changes are also noted in highlighting.
RESPONSE TO REVIEWER 1
Reviewer’s comment:
In this work, bioactive graphene/quantum dots polymer composites for biomedical applications are reviewed.
I would like to address the following comments to the Authors:
Comment 1: The originality and novelty of this review are unclear. What makes this review stand out from other recent reviews on the topic? This must be highlighted.
Author’s response:
I appreciate your consideration. This review article mentioned recent bioactive graphene quantum dots, polymer composites, and their biomedical application. After revision, we have also been updated with current approaches for synthesizing bioactive GQD, bioactive carbon sources, bioactivation roles, bioactivation design, methods for polymerization, and cytotoxicity and biocompatibility, as well as a future perspective. According to the best of our study, there is no comprehensive review conducted on essential properties of GQD polymerization, especially regarding bioactivation in recent years. Also, the current biomedical application was in our scope in this review.
Comment 2: The aim of the study in the last paragraph of the Introduction indicates that the focus of this paper is on the synthesis methods of bioactive graphene quantum dot-based polymer composites. However, only 1.5 pages of the article are devoted to the synthesis methods, while as many as 7 pages are devoted to the application areas. The aim should therefore be reworded.
Author’s response:
Thank you very much for your excellent and precise comment. It is true. According to your value statement, we thoroughly revised the manuscript and reviewed different methods for bioactivation and polymerization of GQD. The following sections were added to the manuscript and highlighted in the text:
Hydrothermal method [69-71], microwave method [72-74], electrochemical oxidation method [75-77], the ultrasonic method [75, 78], and biosynthesis methods [79, 80] are among the methods for bioactive GQDs synthesis based on top-down and bottom-up processes.
2.5.1. Hydrothermal method
A simple and fast way to prepare bioactive GQDs is the hydrothermal method. The hydrothermal method is a one-step method in which the organic precursor is heated in a Teflon tube to reach high temperatures and pressures. Through high temperatures under high pressure, bonds between carbon materials are broken to form bioactive GQDs. The electrical-optical properties of the particles can be adjusted by using different precursors and temperature optimization. Therefore, the hydrothermal method is low-cost and non-toxic (Figure 3). Hydrothermal methods also produce particles with a diameter of 10 nm, more significant than other dot production methods such as electrochemical methods [120-125]. Lianget al. Easily obtained highly fluorescent quantum dots from gelatin by hydrothermal method [126]. Liu and colleagues developed a simple, green, and inexpensive way to prepare fluorescent quantum dots using hydrothermal processing of a type of radish. They investigated the application of prepared carbon quantum dots in cell imaging and detection of iron ions [127]. Hong Miao et al. Presented a green method for synthesizing carbon quantum dots with a high quantum yield of about 14% of tomato paste and investigated the use of synthesized quantum dots in determining the type of antigen [128].
Figure 3. Synthesis of photoluminescence GQDs by hydrothermal method from material A
2.5.2. Microwave method
The synthesis of nanoparticles by microwave-facilitated methods has many advantages compared to previous methods. Compared to the hydrothermal method, this method has the advantage of higher speed and lower synthesis temperature. Some of the advantages of microwave facilitated reactions are: 1) Very low level of impurities in products, 2) Very high product efficiency, 3) Easy control of temperature and pressure, 4) Easy control of product properties (such as size, 5) Environmental friendliness of the method, 6) very high security of the method, 7) reproducibility and 8) selective heating (reduction of energy costs, etc.) [129-131]. Ayele et al. Have proposed a green method for mass production of CdSe quantum dots by microwaves [132]. An easy, light and fast method for the synthesis of nitrogen-doped GQDs using microwave preparation of cedar tree root without surface modification has been proposed by Shawing Sheng et al. [133]. The resulting quantum dots were used to selectively identify Hg2 + ions. Synthesized GQDs can also be used as multicolor fluorescents in bioimaging. Purification of dialysis membranes for 7 hours and GQDs by microwave irradiation for 10 min with raw materials such as aspartic acid (Asp), NH4HCO3, and solvent (DI water) were investigated by Zhang et al. The results showed that the prepared GQDs had a strong blue fluorescence and a QY of 14% Figure 4 [134].
Figure 4. Purified GQDs by microwave irradiation
2.5.3. Electrochemical oxidation methods
Electrochemical methods make it possible to adjust nanostructures by controlling the current / voltage ratio. For example, applying a controlled voltage to a bulk of carbon precursors leads to electrochemical corrosion reactions of carbon reactants and the production of carbon nanostructures. This method does not require high temperatures; It can be easily done on a large scale with aqueous or non-aqueous solvents. This method is one of the fastest producing graphene sheets Figure 5 [103-105, 135, 136]. For example, Paulo et al. GQDs were prepared using electrochemical and corrosion methods [137]. Wong et al. Proposed a bottom-up approach for synthesizing nitrogen-doped GQDs using the electrochemical method. This method is green and simple and is suitable for large-scale synthesis with an interest rate of more than 95%. The quantum gain of the particles is 0.71. Also, the toxicity of quantum dots synthesized by this method is low [138].
Figure 5. The synthesis method of electrochemical oxidation
2.5.4. Ultrasonic method
In recent years, a large number of methods for making photoactive materials for use in solar cells and photocatalysts have been proposed. However, the benefits of preparing these materials through unusual methods are very interesting from the point of view of green chemistry. The idea of ​​producing highly efficient photocatalysts through ultrasound provides a very interesting and important background in science and technology and has great potential for the production of photocatalysts in the near future. Ultrasonic synthesis is a promising method that allows control over size, morphology, nanostructure, and catalytic properties. Compared to conventional energy sources, the supersonic provides abnormal reaction conditions in the liquid phase by the phenomenon of cavitation (very high temperatures and pressures in concise periods of time in liquids) [139-141]. Zhu et al. Synthesized high-quantum GQDs from only graphene oxide and potassium permanganate in a one-step method using ultrasonic irradiation and used them as a test for the detection of alkaline phosphate (ALP) [142]. Oza et al. Prepared GQD in a fast and green way using the precursor chemical acoustic method of lemon [143]. The resulting quantum dots were purified using the sucrose gradient density centrifugation method, which resulted in the separation of water-soluble, single-spray, photoluminescent, highly stable optical and chemical quantum dots. Also, quantum dots synthesized in this way are not toxic and can be used as optical imaging tools. The formation of tens of thousands of small bubbles in the liquid and the loss of carbon-carbon bonds are generated by mechanical force and eventually turn into GQD cutting, this issue is affected by ultrasound waves Figure 6.
Figure 6. The synthesis method of ultrasonic
2.5.5. Biosynthesis method
The biosynthesis method is performed in mild conditions, but also the bioactive GQDs synthesized by this method have inherent biocompatibility and bio-stability without the need for encapsulation operations. Therefore these methods provide a green way to prepare biocompatible, bioactive GQDs. As with chemical methods, biosynthesized quantum dot emissions can be controlled by size, instantaneous excitation of bioactive GQDs of different sizes by a single light source, and a wide spectrum window that can be adjusted by the constructive component [107, 108, 144, 145].
Comment 3: The description of polymer nanocomposites in the last paragraph on page 2 is not entirely accurate and clear. The definition of polymer nanocomposites (page 2, lines 81-82) does not correlate fully with the EU adopted definition of a nanomaterial (2011/696/EU). In the next sentence, polymer nanocomposites are compared with conventional nanocomposites (page 2, lines 82-84). What does "conventional nanocomposites" mean in this sentence?
Author’s response:
Thanks a lot for your valuable view of point. You are right, and accordingly, we carefully read the EU adopted definition of a nanomaterial (2011/696/EU) again in the website (https://eur-lex.europa.eu/legal-content/EN/TXT/?uri=CELEX:32011H0696 ). We revised the manuscript and highlighted the changes as follow:
The polymers in which fillers (in size range from 1 to 100 nm) are homogeneously reinforced are called polymer nanocomposites (PNCs). In these materials, fillers are specified in at least one dimension less than 100 nm (< 100 nm). Based on the number of dimensions that are outside the nanometer range, nanofillers can be classified into three categories: Spherical (zero-dimensional) 2. layered (one-dimensional) 3. fibrous and tubular (two-dimensional).
What does "conventional nanocomposites" mean in this sentence?
Thank you for your attention. It was “conventional composites,” and we corrected it in the manuscript.
Comment 4: Chapter 2 cites only one literature source from the last 5 years, despite the fact that many articles and even reviews have been published on this topic. I suggest that this chapter be expanded and supplemented with more recent literature.
Author’s response:
Thanks a lot, and you are right. According to your comment, we thoroughly revised and added a different section to the manuscript about synthesis methods, bioactivation, and polymerization. Also, we expand and supplement many articles and reviews recently published on this topic. Additionally, we provide a table that shows different synthesis method in some literature as follow:
|
Synthesis methods |
Application |
Ref. |
|
Bottom-up method |
||
|
mango leaf extract- mGQDs |
NIR-responsive fluorescence bioimaging |
[82] |
|
PEGylated GQD |
Fluorescence imaging of tumor |
[83] |
|
GQDs |
Drug delivery and bioimaging |
[85] |
|
Top-down method |
||
|
durian extract - GQDs |
Bioimaging |
[88] |
|
NP-GQD |
Cysteine detection |
[89] |
|
lignin - GQDs |
Bioimaging |
[91] |
Comment 5: The text of the Conclusions is very poor and general. Recent advances and perspectives are not specified. The meaning of the first two sentences is the same.
Author’s response:
Thank you for your valuable comment. We revised the entire conclusion section. We discussed and summarized the topic in more detail. Also, we provide a perspective of the future in this issue as follow:
In the last few decades, bioactive GQDs have attracted much attention due to their properties and applications in various environmental and health fields. Some problems need to be addressed for their practical biological applications such as preparations of bioactive GQDs, their size, reproducibility and low quantum efficiency. The size of bioactive GQD is responsible for their fluorescence properties. The low toxicity of Bioactive GQDs also makes them suitable for use in many in vivo applications. Therefore, it is possible to expand its potential applications in various fields by overcoming the problem of their low quantum efficiency by preparing Bioactive GQD nanocomposites with surface factorization and bandgap engineering. In this study, bioactive GQDs were introduced, and their synthesis methods were explained. The various biomedical applications of bio-active GQDs such as bioimaging (in vivo and in vitro), drug delivery, gene delivery, photothermal therapy, photodynamic therapy, and tissue engineering were discussed. The effortless combination of many drugs and ligands is provided through the large surface and functional groups on bioactive GQDs. Taking advantage of this, bioactive GQDs can be used as a nanocarrier for targeted drug delivery. Also, photoluminescence of bioactive GQDs is used to develop bioimaging techniques to identify various biomolecules that offer widely new strategies for disease diagnosis. It can be used to image cancer cells and to track drug delivery to cells and tissues. However, bioactive GQDs have little toxicity in the human body while having adjustable optical properties and fluorescence propagation in the NIR region of the light spectrum. Researchers aim to develop bioactive GQDs for bio-medical applications facing a key challenge in achieving high-quality products. Small-scale production of bioactive GQDs with large size distributions is possible through existing synthesis methods. Therefore, it is necessary to find new methods to achieve high performance and easy purification methods that do not require the removal of raw materials. Recent research advances in polymer composites based on bioactive GQDs focusing on their synthesis and biomedical applications, including bioimaging (in vivo and in vitro), drug delivery, gene delivery, light therapy, photodynamic therapy, and tissue engineering, are summarized in this review. Finally, we conclude that there is a promising future for further developing GQD-based polymer composites for many unresolved therapeutic barriers.
Comment 6: There are many spelling and grammar errors in the text.
Author’s response:
The manuscript was fully revised, and, as much as possible, we tried to minimize grammatical and typographical errors in this limited time. We gave the text to an expert in English who is familiar with scientific topics to correct the text.

Reviewer 2 Report
Unify the phont color of all figures, some of them are black and the others with violet color.
In addition, all figures need to be redrawn with high resolution.
The permission for the copyright of figures are necessary before acceptance and publication.
The authors should create a table to collect and compare between all previous works.
all figure frames should be deleted.
Delete the abbreviations from title and keywords.
Author Response
Manuscript Ref. No: polymers-1561996
Bioactive Graphene quantum dots Based Polymer Composite for Biomedical Applications
Dear Editor-in-Chief Journal of Polymers,
Thank you for your letter and the opportunity to revise our manuscript ‘Bioactive Graphene quantum dots Based Polymer Composite for Biomedical Applications’ in the journal Computational and Mathematical Methods in Medicine. We appreciate you and the reviewer for the time and effort dedicated to providing thoughtful comments and constructive suggestions.
Below, the reviewer’s comments and cognate responses are written one by one, indicating exactly how we addressed each concern or problem and describing the changes we have made. The changes are also noted in highlighting.
RESPONSE TO REVIEWER 2
Reviewer’s comment:
Comment 1: Unify the phont color of all figures, some of them are black and the others with violet color.
Author’s response:
Thank you very much for your accurate comment. We unify the font color of all figures and tables. This was due to the separate production of manuscript and figures and their subsequent merging.
Comment 2: In addition, all figures need to be redrawn with high resolution.
Author’s response:
Thank you again. The figures were redesigned and presented with better quality.
Comment 3: The permission for the copyright of figures are necessary before acceptance and publication.
Author’s response:
It must be noted that all figures of this manuscript are original.
Comment 4: The authors should create a table to collect and compare between all previous works.
Author’s response:
Thank you for your wise comment. According to your comment, we thoroughly revised and added a different section to the manuscript about synthesis methods, bioactivation, and polymerization. Also, we provide a table in which compare other synthesis methods, size, and applications as follow:
|
Synthesis methods |
Application |
Size |
Ref. |
|
|
|
Bottom-up method |
|||
|
GQD-RhB-silka |
Diagnosis |
3-20 nm |
[109] |
|
|
mango leaf extract- mGQDs |
NIR-responsive fluorescence bioimaging |
2-8 nm |
[110] |
|
|
PEGylated GQD |
Fluorescence imaging of tumor |
2.75 nm |
[111] |
|
|
GQD-PEI |
Gene transfection |
3–4 nm |
[112] |
|
|
GQDs |
Drug delivery and bioimaging |
∼12 nm |
[113] |
|
|
MSN-SS-CD-DOX |
Targeted and controlled drug |
2.7 nm |
[114] |
|
|
|
Top-down method |
|||
|
GQD |
Diagnosis |
5 nm |
[115] |
|
|
durian extract - GQDs |
Bioimaging |
2-6 nm |
[116] |
|
|
NP-GQD |
Cysteine detection |
10–30 nm |
[117] |
|
|
GQD-PEG-AG |
Radiotherapy |
3–4 nm |
[118] |
|
|
lignin – GQDs |
Bioimaging |
2-6 nm |
[119] |
|
Comment 5: all figure frames should be deleted.
Author’s response:
Thanks a lot. As the figures were redesigned and presented with better quality, all figure frames were deleted. We also can submit figures separately to the journal to ensure good quality if needed.
Comment 6: Delete the abbreviations from title and keywords
Author’s response:
Thank you very much. All abbreviations were removed from the title and keywords accordingly.

Reviewer 3 Report
Not a bad little review but I think it can have some improvements.
The structure of the article should be more organized, with information on how the review is set up.
You have to take into account how the luminescence of carbon quantum dots have been identified as coming from many different mechanisms can you include more details about that in the introduction section.
You should include some more details about the examination of toxicity of these materials. Please look at dosage relation to toxic effects in lab studies and into studies on the release of carbon quantum dots from polymer composites into medium.
Author Response
Manuscript Ref. No: polymers-1561996
Bioactive Graphene quantum dots Based Polymer Composite for Biomedical Applications
Dear Editor-in-Chief Journal of Polymers,
Thank you for your letter and the opportunity to revise our manuscript ‘Bioactive Graphene quantum dots Based Polymer Composite for Biomedical Applications’ in the journal Computational and Mathematical Methods in Medicine. We appreciate you and the reviewer for the time and effort dedicated to providing thoughtful comments and constructive suggestions.
Below, the reviewer’s comments and cognate responses are written one by one, indicating exactly how we addressed each concern or problem and describing the changes we have made. The changes are also noted in highlighting.
RESPONSE TO REVIEWER 3
Reviewer’s comment:
Comment 1: The structure of the article should be more organized, with information on how the review is set up.
Author’s response:
Thank you so much for your consideration. We thoroughly revised the manuscript, and the article's structure was re-organized. Much recent information about synthesis, bioactivation, and polymerization was added to the manuscript.
Comment 2: You have to take into account how the luminescence of carbon quantum dots has been identified as coming from many different mechanisms can you include more details about that in the introduction section.
Author’s response:
Thank you for your valuable comment. According to your comment, we add this part to the Introduction section.
Photoluminescence is one of the main properties of GQD that sometimes explain the diversity of its biomedical applications. The photoluminescence mechanisms of GQD are not well known because their structure is complex, and its surface functionalities are still unknown. Therefore, achieving optimal optical properties that affect biomedical applications is challenging.
So far, no mechanism could comprehensively explain the GQDs photoluminescence phenomenon, limiting the regulation of its optical properties. However, several mechanisms have been proposed, including quantum confinement effect, emission of surface states, molecular fluorophores, molecular diffusion of polycyclic aromatic hydrocarbons, self-trapped exciton emission, surface dipole emission center, aggregate emission center, multiple dissolution center, relaxation of slowed solvent center and solvatochromic shift. Each mechanism only partially explains the photoluminescence phenomenon of GQDs. It has been suggested that bioactive compounds can specifically improve photoluminescence. Polymerization of GQDs structures can also increase the biological effects of GQDs.
Comment 3: You should include some more details about the examination of toxicity of these materials. Please look at dosage relation to toxic effects in lab studies and into studies on the release of carbon quantum dots from polymer composites into the medium.
Author’s response:
You are right. It is necessary to pay attention to the biocompatibility of nanostructures. According to your comment, we added a separate section, namely “Cytotoxicity and biocompatibility,” as follow:
Cytotoxicity and biocompatibility
Size and shape control, doping of non-metallic ions, and modification of graphene surfaces and edges by photoluminescence (PL) GQDs, one of the most prominent properties of graphene, are easily adjustable [289, 290]. Large surface area, easy surface modification, high photostability, low cytotoxicity, and excellent biocompatibility, besides adjustable PL, are the features of GQDs [291, 292]. Compared to conventional semiconductor QDs and other carbon-based luminescent nanomaterials, photoluminescence GQDs are more durable. GQDs for biomedical applications due to adjustable PL, excellent biocompatibility, low cytotoxicity, high biocompatibility, and ease of functionalization, as an imaging and labeling luminescent agent are attractive for tracking molecular targets in living cells or organs [293].

Round 2
Reviewer 1 Report
The manuscript was corrected according to the reviewer's remarks. However, the text of the highlights should be rewritten.
Author Response
Manuscript Ref. No.: 1561996
Bioactive Graphene quantum dots Based Polymer Composite for Biomedical Applications
Dear Editor-in-Chief Journal of Polymers,
Thank you for your second letter and the opportunity to revise our manuscript ‘Bioactive Graphene quantum dots Based Polymer Composite for Biomedical Applications’ again for the journal “Polymers”. We appreciate you and the reviewer for the time and effort dedicated to providing thoughtful comments and constructive suggestions.
Below, the reviewer’s comments and cognate responses are written one by one, indicating exactly how we addressed each concern or problem and describing the changes we have made. The changes are also noted in highlighting.
RESPONSE TO REVIEWER 1
Reviewer’s comment:
Comment: The manuscript was corrected according to the reviewer's remarks. However, the text of the highlights should be rewritten.
Author’s response:
We thank you for your valuable comments. These highlights were prepared in response to the previous comment of reviewer No.1. When we reviewed the journal format and previous journal articles, we realized that there was no need for a separate section as highlights. However, considering the importance and value of the comments of the esteemed reviewer, we revised this section as follows:
Highlights
- This study aimed to introduce the essential properties of graphene quantum dots polymerization, especially in the case of bioactivation.
- Different methods for synthesis, bioactivation, and polymerization of graphene quantum dots were reviewed.
- The applications of graphene quantum dots polymerization in biomedicine, including drug delivery, gene delivery, thermal therapy, thermodynamic therapy, bioimaging, and tissue engineering, were discussed.
- Also, the potential cytotoxicity and biocompatibility and future perspectives of bioactive graphene quantum dots-based polymer composites were mentioned.
We sincerely thank you again for taking the time and effort to evaluate our manuscript and hope that this manuscript has found the ability to publish in the valuable journal “Polymers” in the version.
Prof. Wei-Hung Chiang
On behalf of all authors
